Review Article

EMBO
reports

# Interplay between the brain and adipose tissue: a metabolic conversation

Francisco Díaz-Castro [iD] [1,2,3], Eugenia Morselli [iD] [2✉] & Marc Claret [iD] [1,4,5✉]

## Abstract

The central nervous system and adipose tissue interact through complex communication. This bidirectional signaling regulates metabolic functions. The hypothalamus, a key homeostatic brain region, integrates exteroceptive and interoceptive signals to control appetite, energy expenditure, glucose, and lipid metabolism. This regulation is partly achieved via the nervous modulation of white (WAT) and brown (BAT) adipose tissue. In this review, we highlight the roles of sympathetic and parasympathetic innervation in regulating WAT and BAT activities, such as lipolysis and thermogenesis. Adipose tissue, in turn, plays a dual role as an energy reservoir and an endocrine organ, secreting hormones that influence brain function and metabolic health. In addition, this review focuses on recently uncovered communication pathways, including extracellular vesicles and neuro-mesenchymal units, which add new layers of regulation and complexity to the brain–adipose tissue interaction. Finally, we also examine the consequences of disrupted communication between the brain and adipose tissue in metabolic disorders like obesity and type-2 diabetes, emphasizing the potential for new therapeutic strategies targeting these pathways to improve metabolic health.

**Keywords** Brain; Brown Adipose Tissue; White Adipose Tissue; Sympathetic and Parasympathetic Innervation; Adipokines
**Subject Categories** Metabolism; Neuroscience

## Introduction

The central nervous system (CNS) can be regarded as a sophisticated interface computer. It senses and integrates a variety of signals from the body's internal milieu and the environment, executing multiple biological programs to maintain optimal health and ensure survival. This is achieved by modulating essential functions such as appetite, energy expenditure, metabolism, and behavior. While these functions are modulated by complex and distributed neurocircuits across multiple brain regions, the hypothalamus (particularly the arcuate nucleus; ARC)

critically contributes to these biological processes. This occurs in part through the coordinated action of the melanocortin system, specifically pro-opiomelanocortin (POMC) and agouti-related peptide (AgRP)-expressing neurons (Brüning and Fenselau, 2023). Therefore, adequate organismal homeostasis relies on intricate multiway communication routes between organs, with the brain acting as a central processing unit.

Adipose tissue is a fundamental and dynamic organ found across a wide range of the animal kingdom, serving as a buffer for excess calories, a metabolic hub, and an endocrine platform. Adipose tissue comprises two types of depots: white adipose tissue (WAT) and brown adipose tissue (BAT), each exhibiting unique anatomical, functional, and metabolic characteristics crucial for energy balance and physiological regulation. WAT is the predominant type of adipose tissue, functioning as an insulator, energy reservoir (in the form of unilocular storage of triglycerides), and endocrine organ by the release of adipokines (e.g., leptin, adiponectin, resistin) and other signaling molecules (e.g., cytokines, fatty acids, exosomes). In contrast, BAT is selectively located in specific regions, exhibits abundant mitochondria, and specializes in thermogenesis via the distinct expression of uncoupling protein 1 (UCP1). Therefore, precise control of adipose tissue function is key for adapting to the ever-changing metabolic demands and maintaining metabolic health.

In this review, we delve into the bidirectional communication between the CNS and adipose tissue. We examine the anatomical nature, functions, and integration features of efferent (from brain to adipose tissue) and afferent (from adipose tissue to brain) signals upon adipose tissue physiology and systemic metabolism. We also explore novel modes of communication between the brain and adipose tissue, beyond classic nervous innervation, with implications for metabolic control. Finally, we also highlight the pathophysiological consequences of brain–adipose tissue axis dysfunction in the context of prevalent metabolic conditions such as obesity and type-2 diabetes (T2D).

## Brain → white adipose tissue crosstalk

### Efferent innervation

#### Anatomical and functional aspects of sympathetic innervation to the white adipose tissue

WAT is directly innervated by sympathetic endings of the autonomic nervous system (Fig. 1). In response to norepinephrine

[1]Neuronal Control of Metabolism (NeuCoMe) Laboratory, Institut d'Investigacions Biomèdiques August Pi i Sunyer (IDIBAPS), Barcelona, Spain. [2]Laboratory of Autophagy and Metabolism, Faculty of Medicine and Sciences, Department of Basic Sciences, Universidad San Sebastián, Santiago de Chile, Chile. [3]Physiology Department, Biological Science Faculty, Pontificia Universidad Católica de Chile, Santiago de Chile, Chile. [4]IBER de Diabetes y Enfermedades Metabólicas Asociadas (CIBERDEM), Barcelona, Spain. [5]School of Medicine, Universitat de Barcelona, Barcelona, Spain. ✉E-mail: eugenia.morselli@uss.cl; mclaret@recerca.clinic.cat

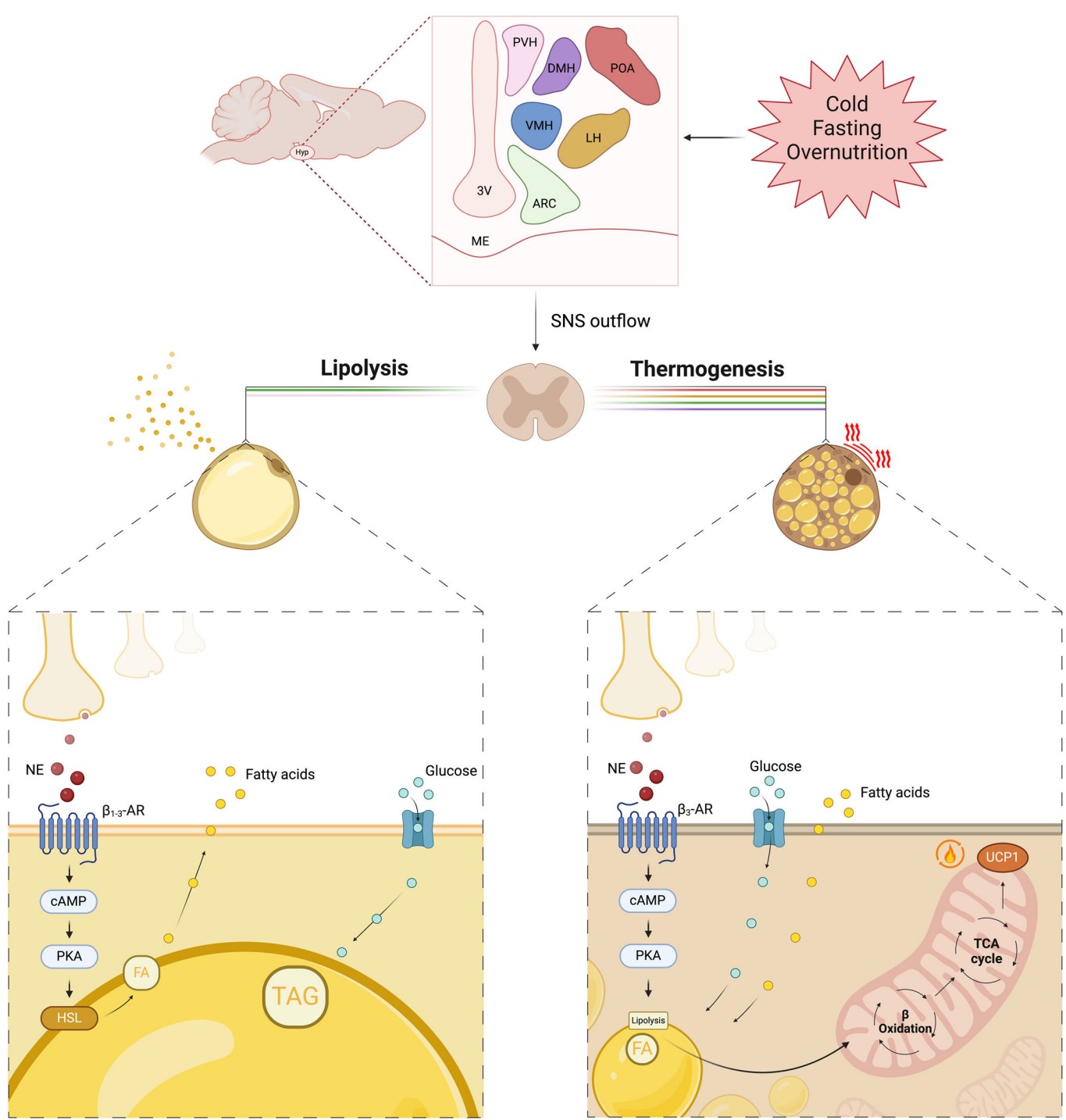

**Figure 1. Neural regulation of adipocyte function by the sympathetic nervous system.**

The role of the sympathetic nervous system (SNS) in regulating adipose tissue function in response to metabolic challenges is illustrated. The upper panel illustrates the brain regions involved in this process in different colors, including the paraventricular nucleus (PVH), dorsomedial nucleus (DMH), ventromedial nucleus (VMH), lateral hypothalamus (LH), arcuate nucleus (ARC), and pre-optical area (POA). The lower panels detail the specific actions of SNS outflow on white adipose tissue (WAT) (left) and brown adipose tissue (BAT) (right) referring to the color of the different hypothalamic nuclei related to lipolysis and thermogenesis. In WAT, norepinephrine (NE) binds to β3-adrenergic receptors (β3-AR), triggering a signaling cascade involving cyclic AMP (cAMP) and protein kinase A (PKA) that leads to the activation of hormone-sensitive lipase (HSL) and subsequent lipolysis, releasing fatty acids (FA). In BAT, NE binds to β3-AR, promoting the uptake of glucose and fatty acids. These substrates undergo β-oxidation in the mitochondria, driving thermogenesis through the action of uncoupling protein 1 (UCP1). This thermogenic response helps to dissipate energy as heat, contributing to the maintenance of energy balance.

(NE) binding to β-adrenergic receptors (βARs) on adipocytes, a signaling cascade is triggered that enhances lipolysis and releases free fatty acids (FFA) and glycerol from white adipocytes (Lass et al, 2011) (Fig. 1). Sympathetic nervous system (SNS) innervation of WAT has been demonstrated by different means, including electrophysiology, surgical sympathectomy, and electrical stimulation of sympathetic nerve endings (Nguyen et al, 2014b). Bamshad et al defined the origins of the sympathetic innervation of WAT using trans-synaptic retrograde neural tracer pseudorabies virus (PRV) in a model of Siberian hamsters. Injection of this virus into the WAT-identified neurons localized in the spinal cord, brainstem, midbrain, and forebrain, in addition to different hypothalamic nuclei (Bamshad et al, 1998). Later studies in rats using neurotrophic viruses defined that WAT projections from the ARC were mainly derived from POMC neurons, while projections from AgRP were absent (Adler et al, 2012). Importantly, WAT is comprised of different fat pads (i.e., subcutaneous versus visceral). Each one of these fat pads is characterized by a distinct innervation density, specific receptor levels, and sympathetic nerve activity (Willows et al, 2023a; Vaughan et al, 2014). Interestingly, sexual dimorphic anatomical differences were also reported in sympathetic WAT innervation (Adler et al, 2012), indicative of a sex-dependent metabolic regulation of WAT. This observation was also confirmed by advanced WAT clearing techniques, which allowed imaging of entire fat pads, revealing that almost 99% of presynaptic fibers within WAT are likely sympathetic (tyrosine hydroxylase (TH) and synaptophysin positive) (Jiang et al, 2017).

A crucial function of sympathetic innervation of WAT is controlling lipolysis. Pioneering in vitro studies performed in the sixties by James W. Correll demonstrated that the concentration of FFA in the bathing medium increased when the nerves of a dissected epididymal fat pad were electrically stimulated (Correll, 1963). This observation was confirmed in different animal models, concluding that surgical or chemical WAT denervation blocked lipid mobilization (Harris, 2018). Importantly, sympathetic innervation is also involved in the development and maintenance of WAT by controlling the entire lifecycle of pre-adipocytes, from their proliferation and differentiation to apoptosis (Cousin et al, 1993; Ruschke et al, 2009; Bowers et al, 2004).

### Anatomical and functional aspects of parasympathetic innervation to the white adipose tissue

The paraventricular nucleus of the hypothalamus (PVH) receives inputs from various hypothalamic nuclei, including the suprachiasmatic nucleus (SCN), which serves as the brain's circadian clock, and the ARC. The PVH sends autonomic projections to the dorsal motor nucleus of the vagus (DMV), where parasympathetic motor neurons that innervate WAT are located. In addition, the PVH projects to the intermediolateral column, the source of sympathetic innervation to WAT.

In contrast to the well-established sympathetic innervation of WAT, which has been confirmed by diverse research groups and study models, the functional outcome of parasympathetic nervous system (PNS) innervation of WAT remains controversial, with different studies reporting conflicting results. Indeed, Giordano et al did not detect neurochemical markers of parasympathetic innervation in various WAT depots using models such as Sprague–Dawley rats, C57Bl/6 mice, and leptin-deficient *ob/ob* mice (Giordano et al, 2006; Bartness et al, 2014). Nonetheless, Kreier et al, in rats, showed that PRV inoculation of intra-

abdominal WAT labeled the DMV nerve, providing evidence of parasympathetic innervation of WAT (Kreier et al, 2002). However, this parasympathetic innervation does not appear to be involved in lipogenesis, which would have been expected as the opposing effect to the lipolytic function of the SNS. Instead, Kreier et al showed that parasympathetic innervation of WAT is implicated in the production of leptin and resistin as well as in insulin sensitivity, as evidenced by the decrease in insulin-dependent glucose uptake and FFA uptake following vagotomy (Kreier et al, 2002).

## Metabolic effects via efferent innervation

As previously mentioned, the SNS exerts significant influence over WAT lipolysis primarily through NE-mediated activation of βARs (subtypes β1-3). Activation of βAR β1-3 engages the GTP-binding protein Gsα, promoting cAMP production via adenylyl cyclase. cAMP enhances protein kinase A (PKA) activity, phosphorylating and activating two crucial proteins for lipolysis: the hormone-sensitive lipase (HSL) and perilipin A (Lass et al, 2011) (Fig. 1). During stress or exercise, the release of NE from sympathetic nerve terminals increases thereby stimulating βARs and promoting FFA release (Santos et al, 2022). βAR levels can be controlled by the CNS as central melanin-concentrating hormone (MCH) infusion, acting on ARC neurons expressing the MCH receptor, diminishes the expression of βARs in WAT (Imbernon et al, 2013).

POMC and AgRP neurons influence WAT lipolysis. Specific activation of AgRP neurons results in reduced circulating fatty acids and activation of HSL in WAT via the SNS (Cavalcanti-de-Albuquerque et al, 2019). Chemogenetic manipulation of POMC neurons curtails the molecular lipolytic program and decreases fasting-induced lipid mobilization (Gómez-Valadés et al, 2021). Leptin, a key adipokine crucially implicated in energy balance control, also exerts a lipolytic effect by activating sympathetic neurons that innervate WAT adipocytes. This effect is partially mediated by the activation of βARs (Zeng et al, 2015). Leptin, by acting on its receptors in AgRP and POMC neurons in the ARC, controls the sympathetic innervation of adipose tissue via a top-down neural pathway that depends on the generation of Brain-Derived Neurotrophic Factor (BDNF) produced by neurons in the PVH (Wang et al, 2020a). Nonetheless, leptin also exerts a concomitant anti-lipogenesis action via hypothalamic phosphoinositide 3-kinase signaling and the endocannabinoid anandamide in WAT (Buettner et al, 2008). Conversely to leptin, insulin infusion into the mediobasal hypothalamus (MBH) dampens SNS activity and enhances WAT lipogenic protein expression, inhibiting HSL and, therefore, lipolysis (Scherer et al, 2011).

Interestingly, it has been reported that the brain controls leptin gene expression in the WAT through sympathetic innervation (Kalsbeek et al, 2001). Indeed, lesions in the SCN, which abolish the biological clock, eliminate the daily rhythm of leptin release (Kalsbeek et al, 2001). Furthermore, SNS innervation also controls the lifecycle (proliferation) of pre-adipocytes. This is evidenced by models of denervation or studies in transgenic mice lacking the neuron-specific Nscl-2 (Nescient Helix–Loop–Helix 2) transcription factor, which exhibit adulthood obesity due to an increased number of pre-adipocytes (Cousin et al, 1993; Ruschke et al, 2009). In Nscl-2 mutant mice, the reduced innervation also leads to a significant decrease in the microvasculature, which might contribute to the observed alterations in WAT (Ruschke et al, 2009). Altogether, these studies reveal the diverse functions of the

Box 1: In need of answers

1. Does the PNS innervate WAT? And, if so, what is its functional outcome?
2. How do sensory nerves and their secreted neuropeptides exert their effects to counteract sympathetic signaling in adipose tissues?
3. What is the precise role of leptin in thermogenesis?
4. How do sensory pathways mechanistically interact with the SNS?
5. Could mechanical forces in WAT or BAT control adipokine/batokine release and represent a new mechanism of adipose tissue →brain communication?
6. How do BAT-derived factors influence brain function and overall energy homeostasis?
7. What is the relevance of bidirectional brain–adipose tissue communication in humans in the context of metabolic control and health?

SNS, ranging from its classical role in the induction of lipolysis to its lesser-studied involvement in hormone production and adipogenesis (Box 1).

In response to external stimuli, such as cold exposure, adrenaline-mediated SNS activation promotes the "browning" of white adipocytes. During this process, white fat cells acquire a brown fat-like phenotype with the primary function of dissipating energy as heat via thermogenesis (Schirinzi et al, 2023). White fat browning is a complex process that is activated by different signaling pathways, which culminates with the expression of UCP1, together with other brown-tissue specific genes (e.g., PPARγ, PRDM-16, and PGC-1α) (Seale et al, 2007). The excitation of the SNS and activation of the β3-AR signaling pathway induced by PRDM-16 is the classic mechanism driving the browning of white adipocytes (Qiang et al, 2012; Seale et al, 2011). Recent studies also indicate that thyroid hormones play a significant role in promoting the browning of WAT, nonetheless this process does not necessarily depend on the SNS (Johann et al, 2019). Taken together, these studies emphasize the importance of innervation in maintaining adipose metabolic health across various levels, including functional, developmental, and lineage aspects.

# Brain → brown adipose tissue crosstalk

BAT is the primary source of nonshivering thermogenesis, capable of increasing energy expenditure by approximately 5-10%, oxidizing intracellular triglycerides, and enhancing glucose uptake upon activation (Celi et al, 2010; Yoneshiro et al, 2011; Orava et al, 2011; Chen et al, 2013; Ouellet et al, 2012). This makes BAT an attractive target to counteract metabolic disturbances. Brown adipocytes, characterized by their abundant mitochondria and multilocular lipid droplets, play a crucial role in adaptive thermogenesis through the expression of UCP1. The interaction between the brain and BAT enables an effective response to challenges such as fasting, cold exposure, and energy balance deregulation.

## Efferent innervation

The communication between the brain and BAT primarily occurs through the SNS (Fig. 1). The intermediolateral nucleus (IML), located between the dorsal and ventral horns of the spinal cord and extending from segment C8 to L1, contains preganglionic sympathetic neurons. This branch is the principal efferent innervation to the BAT (Morrison and Nakamura, 2019) and, like WAT, is predominantly composed of TH cells (Murano et al, 2009). Nervous activation of BAT occurs in response to various stimuli, such as cold exposure, thyroid hormones or energy balance deregulation, activating the SNS. Cold exposure and energy balance deregulation are sensed and integrated by the brain, triggering the release of NE from TH cells leading to the activation of β3-AR. These receptors are abundantly expressed in BAT and play crucial roles in adaptive thermogenesis (Bartness et al, 2010), and their sensitivity is controlled, among other factors, by thyroid hormones. Indeed, central triiodothyronine (T3) stimulates lipid oxidation and therefore a thermogenic program in BAT by acting on AMP-activated protein kinase (AMPK) (Martínez-Sánchez et al, 2017).

In addition, at the intracellular level, and similar to WAT, the binding of NE to β3-AR increases cAMP levels. This second messenger activates PKA, which in turn regulates the activity of HSL. Consequently, this process releases fatty acids, serving as oxidative substrates that stimulate the activation of mitochondrial UCP1 (Cannon and Nedergaard, 2004; Matthias et al, 2000) (Fig. 1). UCP1, embedded in the inner mitochondrial membrane of brown adipocytes, uncouples oxidative phosphorylation from ATP synthesis, allowing protons to flow back into the mitochondrial matrix without generating ATP. Increased UCP1 expression requires T3, and T3-dependent thyroid hormone receptor (TR) β activation (Ribeiro et al, 2010; Bianco and Silva, 1987). UCP1 uncoupling, which characterizes BAT, results in the dissipation of energy as heat that is crucial for maintaining body temperature and energy balance homeostasis (Cannon and Nedergaard, 2004).

The SNS activity involved in BAT thermogenesis originates from the hypothalamus (Fig. 1). Within this region, various nuclei converge to orchestrate the regulation of body temperature and energy balance, aligning with the pivotal role of thermoregulation in metabolic homeostasis. Among the brain regions implicated in the regulation of core body temperature, also known as the "thermoregulatory pathway", the preoptic area (POA) stands out as a crucial nucleus (Morrison and Nakamura, 2019). The POA coordinates a sophisticated circuit that, in a simplified view, responds to low temperatures by receiving inputs from skin thermoreceptors (Nakamura and Morrison, 2008). Subsequently, it generates outputs to other hypothalamic nuclei, such as the dorsomedial hypothalamus (DMH), thus triggering heat production by BAT and contributing to temperature homeostasis (Contreras et al, 2017). Conversely, DMH cholinergic neurons are active in warm temperatures, and their activity is related to the regulation of BAT thermogenesis (Jeong et al, 2015). Lesions in the ventromedial hypothalamus (VMH) have been shown to impair the ability to respond to external cooling or direct stimulation of the POA to induce BAT thermogenesis (Hogan et al, 1982; Preston et al, 1989). These findings underscore the critical role of complex circuits between the POA, VMH, and DMH in body thermoregulation. Among the extra-hypothalamic regions involved in efferent signaling to BAT, the dorsal raphe nucleus (DRN) has been reported as a key player. Electrophysiological recordings have identified certain GABAergic DRN neurons whose firing rates are influenced by fluctuations in local or ambient temperature (Weiss and Aghajanian, 1971; Hale et al, 2011). Moreover, electrical stimulation or pharmacological manipulation of these neurons can alter BAT thermogenesis and concomitantly core temperature (Dib et al, 1994; Higgins et al, 1988). Recently, it has been reported that

GABAergic neurons in the DRN are activated by ambient heat and regulate energy expenditure bidirectionally through changes in both thermogenesis and locomotion (Schneeberger et al, 2019). Optogenetic studies revealed that these neurons influence thermogenesis via direct innervation of the BAT through to the raphe pallidus (RPa) and indirect ascending pathways to the hypothalamus and extended amygdala (Schneeberger et al, 2019). Additional extra-hypothalamic areas have been proposed in thermoregulatory control, such as the inferior olivary nucleus (Uno and Shibata, 2001), highlighting the great complexity of thermoregulation mechanisms.

## Metabolic effects via efferent innervation

During fluctuations in energy balance, as a mechanism to adjust to metabolic demands and maintain overall homeostasis, several hypothalamic nuclei (including the ARC, VMH, and PVH) play key roles in modulating thermogenesis through BAT activation (Contreras et al, 2017). Nutrient levels and hormonal flux, which inform on metabolic status, influence neuronal activity within these nuclei. Importantly, their thermogenic actions can occur independently of "thermoregulatory pathway" circuits (Zheng et al, 1995; Contreras et al, 2017), emphasizing the significance of BAT thermogenic activity in energy homeostasis.

Melanocortin neurons predominantly project to the PVH (Wang et al, 2015), where the release of the POMC-derived neuropeptide alpha-melanocyte-stimulating hormone (α-MSH) binds and activates melanocortin receptors (MCR; mainly MC4-R). This activation is counteracted by the inverse agonist AgRP, which inhibits MCRs and opposes the effects of α-MSH (Nijenhuis et al, 2001). These antagonistic interactions form a critical component of the melanocortin system (De Solis et al, 2024). Through transneuronal retrograde tracing studies using PRVs, it has been demonstrated that SNS connections to the BAT originate from brain areas with notable MC4-R expression (Song et al, 2008). Central pharmacologic interventions with a MCR agonist, such as Melanotan II (MTII), directly into the PVH induces BAT thermogenesis (Monge-Roffarello et al, 2014). Consistent with these findings, several mouse mutants of components of the melanocortin system or direct manipulations of POMC and/or AgRP neurons through optogenetic or chemogenetic activation result in BAT thermogenesis or modulation of BAT glucose uptake, profoundly affecting energy homeostasis (Yu et al, 2020; Schneeberger et al, 2013; Quiñones et al, 2021; Steculorum et al, 2016; Han et al, 2021).

Hypothalamic lipid sensing also modulates BAT activity, with central ceramides emerging as key lipid species influencing sympathetic outflow to BAT. This regulatory mechanism attenuates thermogenesis and promotes a positive energy balance, unveiling a pathway linking lipid signaling in the brain with systemic energy homeostasis (Contreras et al, 2014).

The effects of leptin on BAT thermogenesis are not entirely clear. Several studies using intracerebroventricular (ICV) injections of leptin have shown an increase in BAT and body temperature (Rodríguez-Rodríguez et al, 2019; Pelleymounter et al, 1995), an increase in NE release (Satoh et al, 1999), increase in BAT glucose uptake (Haque et al, 1999), and elevated *Ucp1* mRNA expression in wild-type mice (Rodríguez-Rodríguez et al, 2019; Mistry et al, 1997; Scarpace and Matheny, 1998; Commins et al, 1999). However,

peripheral administration of leptin elicits marginal or no thermogenic response in wild-type or leptin-deficient *ob/ob* mice (Harris et al, 1998). These discrepancies seem to arise from variations in the administration routes, animal models, methods of measuring thermogenesis, and differences between exogenous and endogenous leptin in its physiological effects (Ottaway et al, 2015). Importantly, evidence suggests that leptin may play a role in maintaining body temperature without activating thermogenesis (Fischer et al, 2016). In support of this, it has been reported that sympathetic innervation to BAT is dispensable for leptin-induced weight loss (Côté et al, 2018). As the precise role of leptin in thermogenesis remains unclear, this question can be further explored elsewhere (Fischer et al, 2020).

The VMH is another hypothalamic nucleus that has been extensively associated with BAT thermogenesis. Specifically, neurons expressing steroidogenic factor 1 (SF1; a transcription factor exclusively expressed in the VMH) play a crucial role in this process. Studies deleting the *SF1* gene or ablating SF1 neurons have demonstrated significant impairments in BAT thermogenesis (Majdic et al, 2002; Kim et al, 2011; Rashid et al, 2023). From the mechanistic perspective, accumulating evidence indicates that the energy-sensor AMPK within the VMH may act as a canonical mediator of BAT thermogenesis by integrating diverse homeostatic signals (Seoane-Collazo et al, 2018; Martínez-Sánchez et al, 2017; Beiroa et al, 2014; Martínez de Morentin et al, 2014).

Other systems crucial for the regulation of energy balance, such as the endocannabinoid system, have also been implicated in BAT thermogenesis. Through its interaction with cannabinoid receptors 1 and 2 (CB1R and CB2R), this system has been proposed as a target for several obesity treatments (Miralpeix et al, 2021). Direct administration of the cannabinoid receptor agonist delta-9-tetrahydrocannabinol (D9-THC) has been shown to reduce UCP1 on BAT (Verty et al, 2011), while CB1 antagonist rimonabant increases thermogenesis (Verty et al, 2009). Genetic manipulation of cannabinoid receptors also affects BAT activity (Cardinal et al, 2012).

Most of the information discussed in this section was derived from preclinical models; however, it is important to note that the presence of functional BAT in adult humans was confirmed about 15 years ago. In humans, higher BAT activity positively correlates with a lower body mass index (BMI), younger age, colder outdoor temperatures, female sex, reduced glucose levels, and improved cardiometabolic health (Cohade et al, 2003; van Marken Lichtenbelt et al, 2009; Virtanen et al, 2009; Cypess et al, 2009). However, the brain→BAT crosstalk in maintaining metabolic homeostasis in humans remains largely unknown.

# White adipose tissue → brain crosstalk

## Sensory innervation

The sensory innervation pathway is a sophisticated network that facilitates communication between peripheral fat depots and the CNS (Fig. 2). Sensory neurons, which are pseudo-unipolar and reside in the dorsal root ganglia (DRG) of the spinal cord, have axons extending peripherally toward tissues and centrally toward the brain via the spinal cord. DRG neurons, which are clusters of sensory neuron cell bodies located adjacent to the spinal cord, serve

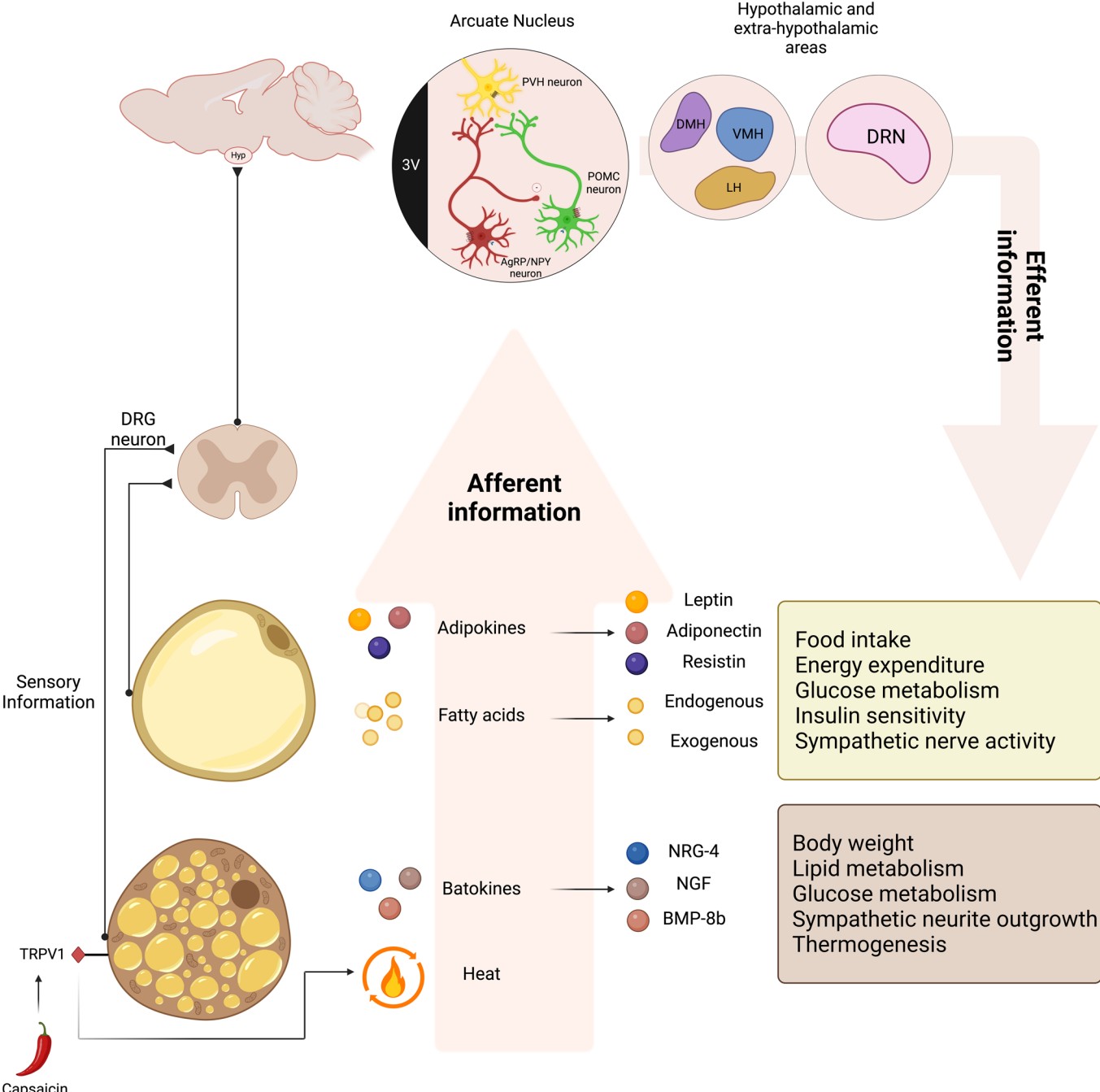

**Figure 2. Afferent neural pathways and adipose tissue-derived signals in energy homeostasis.**

The afferent neural pathways and the signals derived from adipose tissue that contribute to the regulation of energy homeostasis are illustrated. The central panel shows the arcuate nucleus (ARC) in the hypothalamus, which houses first-order neurons, including pro-opiomelanocortin (POMC) and agouti-related peptide/neuropeptide Y (AgRP/NPY) neurons. These neurons detect fluctuations in energy status (afferent information) and, in response, interact with other hypothalamic nuclei, including the dorsomedial hypothalamus (DMH), ventromedial hypothalamus (VMH), lateral hypothalamus (LH), and extra-hypothalamic regions such as the dorsal raphe nucleus (DRN). Collectively, these structures modulate energy balance and metabolic processes (efferent information) to maintain systemic energy homeostasis. Sensory information from adipose tissues is carried by dorsal root ganglion (DRG) neurons to deliver information to the central nervous system. The lower panels detail the specific signals emanating from white adipose tissue (WAT) and brown adipose tissue (BAT). WAT releases adipokines (e.g., leptin, adiponectin, resistin) and fatty acids, which modulate various physiological processes such as food intake, energy expenditure, glucose metabolism, insulin sensitivity, and sympathetic nerve activity. BAT produces batokines (e.g., NRG-4, NGF, BMP-8b) and heat through thermogenesis. These signals influence body weight, lipid metabolism, glucose metabolism, sympathetic neurite outgrowth, and energy balance. The transient receptor potential vanilloid 1 (TRPV1) channels, activated by capsaicin, play a role in transmitting thermal and metabolic information from BAT, thereby integrating sensory information with energy homeostasis. This intricate system of afferent and efferent signaling pathways underscores the complexity of energy regulation mechanisms and highlights the pivotal role of both central and peripheral signals in maintaining metabolic stability.

as a processing hub, relaying sensory information to the brain. The specialized structural and functional adaptations of sensory neurons at their termini allow them to detect a wide range of stimuli through various types of receptors: mechanoreceptors, thermoreceptors, nociceptors, electromagnetic receptors, and chemoreceptors (Valencia-Montoya et al, 2024). Although their function and innervation patterns are still largely unknown, reports have demonstrated sensory afferent pathways originating from the WAT to the brain. These pathways are crucial for regulating lipid mobilization and body fat stores as they transmit information about the metabolic state of fat through hormones secreted by the adipose tissue (Willows et al, 2021; Zhang et al, 2018).

Early immunocytochemistry studies using antibodies against the neuropeptides substance P and CGRP (calcitonin gene-related peptide) (Giordano et al, 1996) or against ubiquitous proteins characteristic of sensory neurons (advillin and Nav18) (Willows et al, 2021) identified the presence of sensory-related innervation in the WAT. This was confirmed using an anterograde transneuronal viral tract tracer, as shown by Song et al, who identified a sensory WAT pathway to the brain in the inguinal and epididymal WAT of the Siberian hamster model (Song et al, 2009). The sensory innervation is mainly composed of "C" fibers, small and non-myelinated, although myelinated fibers have also been described (Giordano et al, 1996; Willows et al, 2021, 2023b). The CNS projections of WAT afferents overlap with SNS outflow sites, creating a WAT sensory-SNS circuit involved in the control of lipid mobilization (Song et al, 2009). More recent work also reported the existence of somatosensory fibers, originating from the DRG, innervating adipose tissue in rodents (Bartness et al, 2014; Fishman and Dark, 1987). These fibers provide fast neural transmission of information from peripheral organs to the brain. It remains unclear which DRG subtypes innervate WAT and how the sensory pathways mechanistically interact with the SNS. Nevertheless, it is well-established that sensory ablation affects fatty acid and lipid metabolism. In addition, sensory ablation increases cold-induced thermogenesis driving the browning of WAT and rising body temperature (Wang et al, 2022b). Sympathetic neuron deletion increases the size of the WAT fat pad, indicating that sensory innervation also counteracts sympathetic activity (Wang et al, 2022b). As such, the afferent flow of sensory information from a WAT pad directly modulates the activity of the sympathetic efferent pathways that innervate the same pad. Altogether, these studies demonstrate a bidirectional neural communication between the brain and WAT, which is relevant for energy balance and metabolic control.

## Endocrine regulation of the adipose–brain axis (adipokines)

As previously mentioned, WAT is a crucial endocrine organ that secretes hormones and other factors collectively known as adipokines. These adipokines play a major role in the regulation of organismal metabolism by informing the brain about whole-body long-term energy-storage status (Fig. 2). The best-studied adipokine is leptin (Zhang et al, 1994), which is produced by the adipose tissue in proportion to fat stores and informs the brain about energy-storage levels in adipose tissue. The leptin receptor (LepR) is most densely distributed in the ARC (Elmquist et al, 1998). Leptin acts on ARC neurons, promoting satiety and

increasing energy expenditure, thus controlling energy balance (Lavoie et al, 2023). Indeed, leptin activates POMC neurons (Cowley et al, 2001) while inhibiting AgRP neurons (van den Top et al, 2004). It also activates presynaptic GABAergic neurons, thereby decreasing the inhibitory tone of postsynaptic POMC neurons (Vong et al, 2011). Recent work has identified an additional population of ARC neurons expressing LepR: prepro-nociceptin (PNOC)-expressing neurons (Jais et al, 2020). The activation of PNOC neurons by short-term high-fat diet (HFD) feeding is mediated by inhibitory synaptic input to POMC neurons, leading to an increase in food intake (Jais et al, 2020). Leptin inhibits a subpopulation of PNOC neurons (around 80% of neurons); nonetheless, the relevance of these neurons in mediating leptin's metabolic effects is currently unknown. This leptin-mediated brain action plays a crucial role in regulating appetite and glucose homeostasis (Berglund et al, 2012; Campfield et al, 1995), in addition to controlling insulin sensitivity in peripheral tissues via a neurocircuit involving the vagus nerve (German et al, 2009). Importantly, leptin not only modulates the neuronal activity of hypothalamic neurons. The LepR has also been identified in astrocytes, whose activation controls synaptic remodeling and food intake (Kim et al, 2014).

Another important adipocyte-derived hormone is adiponectin (Scherer et al, 1995). Adiponectin activates its receptors, AdipoR1 and AdipoR2, expressed on various nuclei of the MBH (Guillod-Maximin et al, 2009; Kubota et al, 2007), modulating food intake (Kubota et al, 2007; Coope et al, 2008), energy metabolism and body weight (Qi et al, 2004), bone metabolism (Kajimura et al, 2013), and glucose metabolism (Koch et al, 2014). Adiponectin can positively or negatively affect food intake by acting on POMC neurons, an effect that depends on the energy state or glucose concentration in the brain (Suyama et al, 2016). Interestingly, POMC but not neuropeptide Y (NPY) neurons are sensitive to glucose concentration. Indeed, adiponectin also enhances inhibitory postsynaptic currents onto NPY neurons, attenuating their firing rate independently of glucose levels (Suyama et al, 2017). Adiponectin also acts on oxytocin neurons by inhibiting their excitability, suggesting a potential role in influencing energy homeostasis through the control of pituitary hormone secretion (Hoyda et al, 2007). In addition, adiponectin influences hypothalamic-controlled circadian rhythms via AdipoR1-mediated upregulation of the core clock gene *Bmal1*, controlling orexigenic neuropeptide expression and food intake (Tsang et al, 2020). Similarly to leptin, adiponectin also triggers catabolic processes in astrocytes, having a positive impact on nutrient availability in the hypothalamus (Song et al, 2021). These findings suggest that adiponectin plays a crucial role in modulating hypothalamic activity and energy homeostasis through multiple mechanisms modulating the function of neurons and non-neuronal populations. Nonetheless, adiponectin not only acts on the hypothalamus in the brain. Indeed, AdipoR1 and AdipoR2 have also been identified in the nucleus of the solitary tract, where they decrease blood pressure by modulating the excitability of NPY neurons, thus influencing central autonomic processing (Hoyda et al, 2009).

The WAT also communicates with the brain through resistin, an adipokine that is produced in adipocytes in rodents and adipose tissue-infiltrated macrophages in humans, and whose levels correlate with adiposity. Its name derives from its action to induce

resistance to insulin (Steppan et al, 2001); additionally, this adipokine influences metabolic regulation and inflammatory responses by acting on the hypothalamus. Indeed, although a specific resistin receptor has not yet been identified, TLR4 has been recognized as a binding site for resistin, where it activates a TLR4-mediated signaling pathway involved in resistin-induced inflammation and insulin resistance (Benomar et al, 2013). Resistin acts in the brain increasing sympathetic nerve activity, affecting the cardiovascular system, but inhibits sympathetic nerve activity to BAT (Steppan et al, 2001). The literature is still limited, however, crosstalk between leptin and resistin on metabolic function has been described, particularly through their action on the hypothalamus. Evidence indicates that resistin reduces the action of leptin on food intake and has opposing actions on insulin sensitivity. However, the precise mechanisms that underlie this crosstalk are still a matter of investigation.

Mounting evidence has shown that a growing list of adipokines produced by the WAT is involved in energy and metabolic homeostasis through their actions on the brain. Apelin, acting on its receptor (APJ), exerts pleiotropic effects that include glucose homeostasis, control of blood pressure, nutritional behavior, and the secretion of pituitary hormones (Mehri et al, 2023). Nonetheless, it is important to mention that apelin is produced by diverse tissues and cell types, including hypothalamic POMC neurons (Reaux-Le Goazigo et al, 2011), and as such it is difficult to determine the specific impact of WAT-derived apelin in WAT–brain communication. Additional adipokines, such as omentin-1 and retinol-binding protein 4 (RBP4), have been proposed to act on the hypothalamus. However, to date, these hypothalamic effects appear to be secondary to their peripheral actions (Brunetti et al, 2011).

## Metabolic regulation of the adipose–brain axis (fatty acids)

WAT also secretes various species of lipids into the circulation to communicate with the brain (Fig. 2). Fatty acids are released from adipocytes into the bloodstream, often bound to albumin or incorporated as triglycerides within lipoproteins. As they circulate, they can cross the blood-brain barrier (BBB) through diverse transport mechanisms, including passive diffusion or facilitated transport (Pifferi et al, 2021). The BBB has specific transport proteins that help shuttle fatty acids into the brain, among these are fatty acid transport proteins 1–6 (FATP1-6), fatty acid translocase/CD36 (FAT/CD36), intracellular fatty acid binding proteins 1–9 (FABP1-9), plasma membrane fatty acid binding protein (FABPpm), and caveolin-1; for a recent review see (Cruciani-Guglielmacci et al, 2024). More recently, an additional fatty acid transport protein called Mfsd2a has been identified, which specifically binds and transports the long-chain unsaturated fatty acids docosahexaenoic acid (DHA) and α-linolenic acid (ALA), using a unique flipping mechanism (Nguyen et al, 2023; Chua et al, 2023; Nguyen et al, 2014a). Once in the brain, fatty acids can be esterified and/or incorporated into membrane phospholipids (Rapoport et al, 2001). Nonetheless, they can also function as signaling molecules by binding to membrane receptors or by functioning as intracellular messengers modulating signal transduction. One of the first and most studied metabolic functions of fatty acid sensing is the control of energy and glucose homeostasis.

Specialized neurons in the hypothalamus, including AgRP and POMC neurons, detect fluctuations in fatty acid concentration and modulate their activity, thereby orchestrating behavioral and metabolic responses (Cruciani-Guglielmacci et al, 2024). More than two decades ago, the team of Rossetti showed that oleic acid, acting on the brain, inhibits glucose production and food intake (Obici et al, 2002), an effect mediated by FAT/CD36 (Le Foll et al, 2013) and acyl-CoA synthesis (Moullé et al, 2013). More recently, other fatty acid receptors involved in metabolic regulation have been identified in hypothalamic neurons: free fatty acid receptor-4 (FFAR4) and -1 (FFAR1) (Dragano et al, 2017). While FFAR4 has been mainly associated with anti-inflammatory functions (Dragano et al, 2017), FFAR1 in hypothalamic POMC neurons controls food intake and body weight by mechanisms that are still unclear (Dragano et al, 2024). Fatty acid-mediated FFAR1 activation increases intracellular $Ca^{2+}$ levels in hypothalamic neurons, which could affect neuronal activity as well as different cellular processes. Indeed, palmitic acid-mediated activation of FFAR1 in hypothalamic neurons inhibits autophagy, which is considered an important factor in the control of food intake (Hernández-Cáceres et al, 2019; Quan et al, 2012).

Hypothalamic fatty acid sensing also influences the function of several hormones. Fatty acid signaling in the hypothalamus regulates insulin secretion and action, a process dependent on fatty acid β-oxidation and the enzyme carnitine palmitoyl transferase 1 (CPT1) (Migrenne et al, 2006), which regulates the entrance of fatty acids into mitochondria (Obici et al, 2003). Fatty acids in the VMH stimulate the orexigenic effects of ghrelin on food intake, which involves the inhibition of fatty acid biosynthesis induced by AMPK and concomitant activation of CPT1 (López et al, 2008). Fatty acid sensing is also involved in the hepatic secretion of triglyceride-rich lipoproteins (VLDL-TG). Infusion of oleic acid into the MBH activates a PKC-δ → KATP-channel signaling axis, which relies on the dorsal vagal complex (DVC) and hepatic innervation leading to the suppression of VLDL-TG secretion in rats (Yue et al, 2015). Although much of the literature has focused on the role of neurons in lipid sensing, mounting evidence highlights the involvement of hypothalamic astrocytes and tanycytes in fatty acid detection and metabolic actions (Hofmann et al, 2017). However, the mechanisms underlying these processes and whether this information is subsequently transmitted to specific hypothalamic neurons still require further investigation.

## Immune system in the regulation of the brain–adipose axis

The innate immune system plays a critical role in adipose tissue function (Kawai et al, 2021). Among the various immune cells in WAT, macrophages are the most abundant, accounting for more than half of the leukocytes in both lean and obese animals. Their accumulation in WAT is closely associated with obesity (Weisberg et al, 2003). Evidence suggests a bidirectional communication between the CNS and resident immune cells in WAT, influencing lipid homeostasis. Consequently, the immune system may play a key role in mediating the crosstalk between the brain and WAT. This mediation is likely facilitated by the presence of adrenergic receptors on macrophages, specifically β2-AR (Petkevicius et al, 2021) and their ability to regulate sympathetic innervation of WAT (Rahman and Jun, 2022).

Stimulation with a β-adrenergic agonist induces adipose tissue remodeling, triggering a beigeing process that involves adipocyte cell death and their clearance by a coordinated network of non-inflammatory M2 macrophages (Lee et al, 2013, 2016). In addition, cold exposure can induce the secretion of molecules from macrophages that modulate WAT sympathetic innervation. The cytokine Slit3, secreted by M2 macrophages, increases adipose tissue sympathetic activity via the Slit-ROBO pathway, enhancing thermogenesis and increasing the synthesis and release of NE (Wang et al, 2021). BDNF is another molecule secreted by macrophages in response to cold exposure. Studies using various models of BDNF modulation (knockdown and overexpression) (Wang et al, 2020b; Cao et al, 2011, 2009) suggest that BDNF contributes to WAT innervation, potentially through the highly expressed TrkB receptor found in sensory and sympathetic nerves (Blaszkiewicz et al, 2020). In conclusion, while the immune system's role in regulating the brain–adipose axis shows promising potential, our understanding remains limited. Further research is needed to fully uncover these complex interactions and their implications for metabolic regulation.

## Brown adipose tissue → brain crosstalk

### Sensory innervation

Sensory neurons originating in the BAT extend their axons to the DRG (Fig. 2), as anatomical studies in rodents have shown that sensory projections in C4–C8/T1–T2 DRG were traced from BAT (Ryu et al, 2015). Studies using anterograde transneuronal viral tracing have revealed that sensory nerve projections from BAT primarily reach the brainstem, forebrain, and hypothalamus, targeting nuclei that in turn regulate sympathetic output to adipose tissues (Vaughan and Bartness, 2012). WAT exhibits more extensive sensory innervation (Wang et al, 2022b), however, studies that modulate sensory activity in BAT underscored the critical role of these afferent signals. A major focus in sensory nerve research has been on transient receptor potential (TRP) channels, which include receptors from all five types. Among these, TRPV1 has been particularly investigated in a metabolic context due to its capacity to be activated by capsaicin (a compound found in hot chili peppers) (Caterina et al, 1997) (Fig. 2). TRPV1 activation in human randomized controlled trials enhances energy expenditure through BAT thermogenesis and reduces food intake, resulting in a negative energy balance and decreased adiposity (Inoue et al, 2007; Snitker et al, 2009; Yoshioka et al, 2001). In line with this, global double-knockout TRPV1/UCP1 mice show reduced BAT mass and mitochondrial respiration, leading to lower whole-body oxygen consumption and heat production resulting in an obese phenotype (Li et al, 2022). The observed phenotype was more pronounced than the single UCP1 knockout mouse, suggesting a synergistic effect (Li et al, 2022). Together, these results suggest that BAT sensory innervation could be a potential target for anti-obesity drug therapies (Ohyama et al, 2016).

### Endocrine regulation (batokines)

Traditionally, BAT has been studied for its key role in thermogenesis. However, in recent years, the secretory function of BAT has gained significant attention for its impact on interorgan communication (Fig. 2). The secreted molecules, known as "brown adipokines" or "batokines," target distant organs and tissues, including the liver, heart, skeletal muscle, WAT, and brain. They exert their effects through autocrine, paracrine, or endocrine mechanisms (Ahmad et al, 2021). Among these batokines, neuregulin-4 (NRG-4) is highly expressed in BAT compared with WAT (Chen et al, 2017; Christian, 2015) and has been shown to positively influence energy balance, lipid metabolism, and glucose metabolism (Chen et al, 2017; Wang et al, 2014). NRG-4 also promotes sympathetic neurite outgrowth (Pellegrinelli et al, 2018; Rosell et al, 2014), as is also the case for nerve growth factor (NGF) (Néchad et al, 1994; Nisoli et al, 1996).

Another batokine implicated in brain crosstalk is bone morphogenetic protein 8b (BMP-8b), which increases in response to thermogenic and nutritional factors, such as cold exposure or an HFD (Whittle et al, 2012). BMP-8b not only acts autocrinally within the BAT but also plays an endocrine role in the hypothalamus, thereby enhancing BAT thermogenesis and increasing energy expenditure through the SNS (Pellegrinelli et al, 2018; Whittle et al, 2012). BAT also secretes FGF-21, GDF-15, and IL-6, which have been associated with various effects on the brain (Ahmad et al, 2021). However, the specific contribution of these factors when derived from BAT in the context of BAT-brain communication warrants further investigation.

Collectively, these findings emphasize the secretory function of BAT, redefining our understanding of this tissue beyond its role in thermogenesis. This positions BAT as an endocrine organ that influences metabolism through inter-tissue communication. This expanded perspective highlights BAT's potential as a target for therapeutic intervention against metabolic disorders (Ahmad et al, 2021).

## New modes of communication

Emerging research has unveiled novel communication mechanisms between adipose tissue and the brain, influencing physiological processes beyond metabolism and energy balance. In this section, we delve into the nature and diversity of these means of communication, which add another layer of complexity to the intricate interorgan crosstalk.

### Neuro-mesenchymal units

Neuro-mesenchymal units refer to specialized cellular networks that involve communication and interactions between nervous system components and mesenchymal cells. Recent research in adipose tissue biology has highlighted the cooperation between sympathetic neurons and immune cells in maintaining metabolism (Cardoso et al, 2021). Cardoso and collaborators discovered that sympathetic terminals control the expression of glial-derived neurotrophic factor (GDNF) and the activity of type-2 innate lymphoid cells (ILC2s) through β2-ARs in adjacent adipose mesenchymal cells. Disruption of this communication alters the generation of cytokines, thus contributing to metabolic imbalance (Cardoso et al, 2021). This neuroimmune regulation is anatomically and functionally mediated by multiple brain areas, including the PVH which plays a crucial role in systemic metabolism and selectively influences sympathetic output to adipose tissue in a

depot-specific manner (Nguyen et al, 2014b). While our current understanding of neuro-mesenchymal units is still developing, this recent research underscores their relevance as a dynamic communication platform for regulating energy balance.

## Extracellular vesicles

In addition to traditional soluble factors, adipose tissue also secretes extracellular vesicles (EVs). These EVs are lipid bilayer nanoparticles that function as physiological intercellular and interorgan communication vehicles, facilitating the transfer of diverse bioactive cargos (proteins, lipids, diverse RNA species, especially miRNAs, organelle parts/proteins, etc.) that contribute to a wide range of metabolic processes (Isaac et al, 2021). Given the cellular heterogeneity of adipose tissue, EVs can be released by adipocytes, macrophages, adipose stem cells, or endothelial cells (Isaac et al, 2021). The content of EVs is influenced by nutritional or pathophysiological conditions (obesity, T2D) reflecting the metabolic state of the adipose tissue and partially mediating the systemic effects observed in these conditions (Isaac et al, 2021). For example, EVs from obese or diabetic individuals often carry altered molecular signatures that can exacerbate inflammation and insulin resistance, while those from individuals in a fasted or refeed state may promote metabolic adaptations beneficial for maintaining energy homeostasis (Crewe et al, 2018). Numerous reports evidenced EV-mediated communication between the adipose tissue and several organs, such as the pancreas, skeletal muscle, liver, and heart (Crewe and Scherer, 2022). However, much less is known about their interactions with the brain. Gao et al reported that adipocytes from obese mice secrete EVs that can be taken up by POMC-like neurons in vitro (Gao et al, 2020). When these EVs were transferred to lean mice, they promoted appetite and body weight gain. These effects were believed to occur through the transfer of miRNAs and lncRNAs to POMC neurons, thereby stimulating the mTORC1 signaling pathway (Gao et al, 2020). Conversely, EVs from adipocytes of lean mice reduced appetite and mitigated weight gain in mice fed with HFD by decreasing mTORC1 signaling (Gao et al, 2020). Another study demonstrated that extracellular nicotinamide phosphoribosyltransferase (eNAMPT), an enzyme implicated in the biosynthesis of $NAD^+$, is transported in EVs through systemic circulation (Yoshida et al, 2019). The authors found that eNAMPT declined with age (in both mice and humans), and that genetic adipose-specific overexpression of eNAMPT increased $NAD^+$ levels in diverse tissues including the hypothalamus. This was correlated with improved function and antiaging effects in female mice (Yoshida et al, 2019). Interestingly, a specific population of neurons located in the DMH has been shown to influence aging and lifespan via the release of adipose eNAMPT via the SNS (Tokizane et al, 2024). These results suggest a bidirectional brain–adipose tissue communication to maintain $NAD^+$ levels and healthy aging.

Obesity and T2D are associated with cognitive decline (Biessels and Despa, 2018). Control mice treated with adipose tissue-derived EVs from HFD-fed mice or patients with diabetes exhibited a synaptic loss in the hippocampus and cognitive impairments (Wang et al, 2022a). An altered EV miRNA cargo was responsible for these alterations. Specifically, the enrichment of miR-9-3p negatively regulated the expression of BDNF in the hippocampus, which is key for synaptic function (Wang et al, 2022a). Together, these results unveil an EV-mediated communication mode between adipose tissue and the brain implicated in cognitive dysfunction in the context of metabolic disorders.

## Mechanical forces

Recently, research has been focusing on signaling mechanisms beyond nervous and chemical signals in the control of the adipose–brain axis. A hot topic in this area is the importance of mechanical forces, which are currently considered a key factor for the maintenance of WAT health (De Luca et al, 2021). It is tempting to speculate that mechanical forces might control the production and release of adipokines, thereby playing a role in tissue communication.

# Consequences of brain–adipose tissue axis dysfunction

Environmental factors (e.g., imbalanced diets, exposure to toxins, sedentary lifestyle, etc.) can significantly impair organ function by causing metabolic alterations, inflammation, or oxidative stress. These disruptions not only affect individual organs but also the communication routes that coordinate systemic processes, resulting in a cascade of adverse health outcomes (Fig. 3).

WAT is particularly vulnerable to environmental aggressions. A paradigmatic example is the predominant Western lifestyle, characterized by the consumption of obesogenic diets and sedentarism. Energy excess leads to fat accumulation primarily through adipocyte hypertrophy, which is associated with adipose tissue inflammation, fibrosis, and hypoxia, ultimately contributing to poor metabolic health (Sakers et al, 2022). This dysfunctional fat also produces lower levels of adiponectin (an insulin-sensitizing adipokine) and higher levels of leptin (which is produced in proportion to fat stores) (Sakers et al, 2022). In the context of obesity, elevated leptin levels are frequently associated with leptin resistance, a condition marked by the inability to respond to the anorexigenic effects of this hormone. Evidence indicates that hyperleptinemia is required for the development of leptin resistance (Knight et al, 2010), which evolves through the overactivation of leptin signaling in key hypothalamic areas that eventually promote a negative feedback regulation via SOCS-3 (Ernst et al, 2009). This blunts LepR signaling and its appetite-suppressing effects, thus sustaining increased food intake and body weight. Other mechanisms potentially underlying leptin resistance include various molecular and functional alterations associated with structural changes to the molecule, its transport across the BBB, and the deterioration of leptin-receptor function and signaling.

Obesogenic diets also have direct effects on the hypothalamus. Excessive fat and sugar consumption may cause mitochondrial dysfunction (Gonzalez-Franquesa et al, 2022), endoplasmic reticulum (ER) stress (Zhang et al, 2008), autophagy impairment (Meng and Cai, 2011), lipid metabolism deregulation (González-García et al, 2017), and inflammation (De Souza et al, 2005) in the hypothalamus, leading to alterations in neuronal function and contributing to the development of obesity. These pathological processes are closely interrelated and can influence each other in an intricate and synergetic manner. Their combined and sustained effects promote in the hypothalamus accumulation of some types of lipids, such as palmitate or ceramides, enhancing local oxidative stress, the release of cytokines and inflammatory mediators, as well as ER malfunction (Rodriguez-Navas et al, 2016; Contreras et al, 2014). This results in defective neuropeptide expression (e.g., *Npy*, *Agrp*) and processing (e.g., POMC), the deterioration of metabolic signaling pathways (e.g., leptin and insulin),

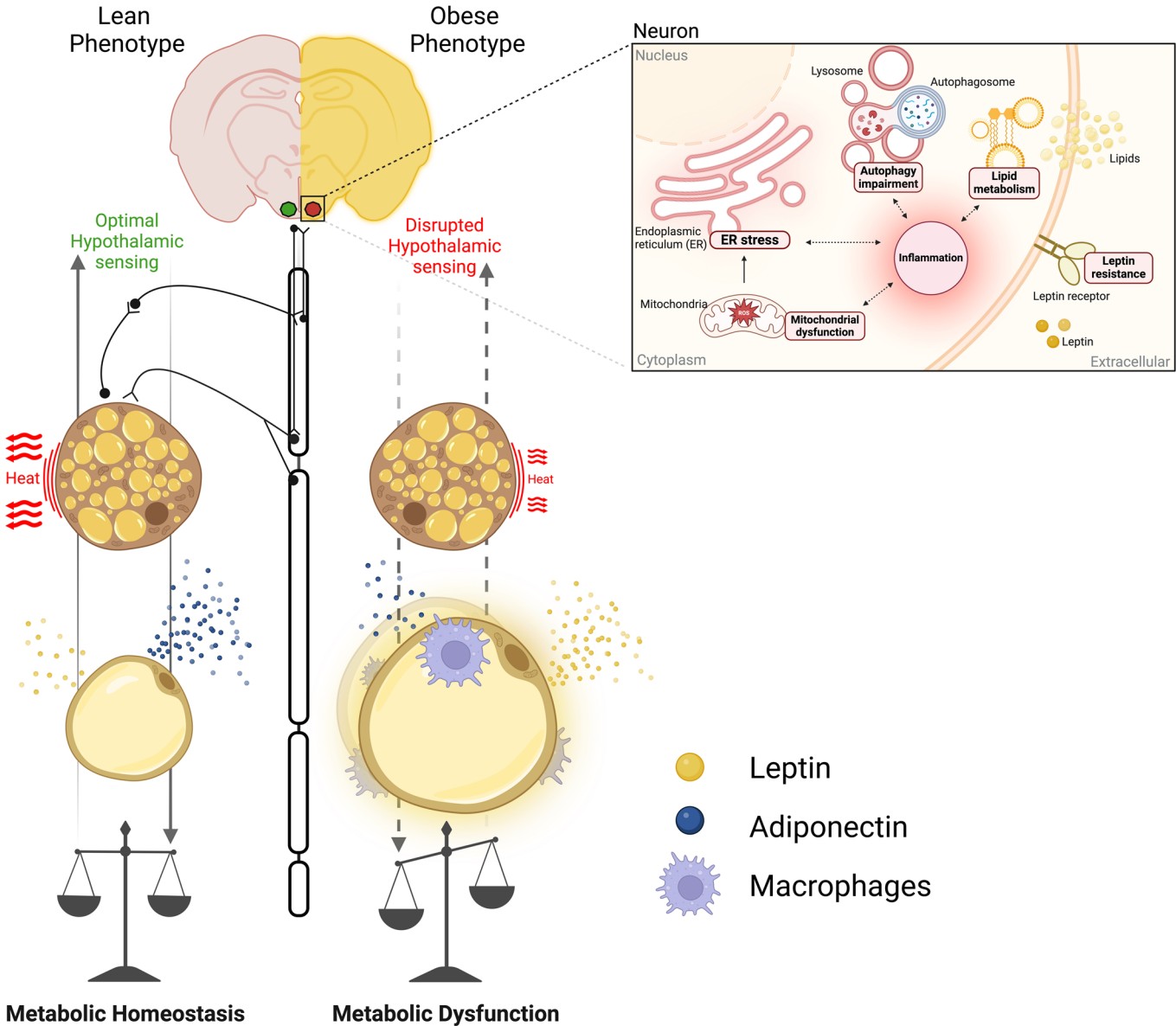

**Figure 3. Brain–Adipose tissue communication in health and obesity.**

The hypothalamic sensing and adipose tissue function between lean and obese phenotypes is illustrated. On the left, the lean phenotype is characterized by optimal hypothalamic sensing, efficient thermogenesis in brown adipose tissue (BAT), and healthy white adipose tissue (WAT) function, converging in systemic a metabolic homeostasis. On the right, the obese phenotype is associated with disrupted hypothalamic sensing, leading to impaired thermogenesis in BAT and dysfunctional WAT. The disrupted sensing results in reduced energy expenditure and altered adipokine secretion, including elevated leptin levels and reduced adiponectin. This condition is also characterized by increased macrophage infiltration into WAT, contributing to chronic inflammation and further metabolic dysregulation. At the neuronal level, the obese phenotype is characterized by heightened inflammation, which may lead to mitochondrial dysfunction, endoplasmic reticulum (ER) stress, disrupted lipid metabolism, and impaired autophagy. These cellular disturbances ultimately contribute to the development of leptin resistance.

and apoptosis of certain neuronal mediators (Ozcan et al, 2009; Cakir et al, 2013; Schneeberger et al, 2013; Williams et al, 2014; Moraes et al, 2009; Yi et al, 2017). Collectively, this promotes increased appetite and altered communication with WAT and BAT, interfering with lipolysis and thermogenesis, and ultimately contributing to the development of obesity. While most of our current understanding of these processes comes from rodent models, evidence suggests that these findings may also be relevant to humans (Lei et al, 2024; Thaler et al, 2012; Kreutzer et al, 2017).

## Conclusions

The bidirectional communication between the brain and adipose tissue is multifaceted and complex, encompassing multiple nervous and hormonal pathways as well as novel and unexpected routes of communication. These multiple layers of signaling enhance the system's ability to fine-tune metabolic processes, integrate diverse metabolic signals, and respond effectively to varying physiological conditions and disruptions, thereby maintaining overall physiological stability and promoting health. While

research on this topic has started to unveil a clearer picture of the interactions involved, there is still much to discover about their nature, molecular mechanisms, functions, and the consequences of their disruption. As research advances, further elucidation of these interactions may lead to novel strategies for managing metabolic diseases and enhancing overall metabolic health.

## Peer review information

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

## Acknowledgements

This work was supported by the Departament de Recerca i Universitats de la Generalitat de Catalunya (2021-SGR-01320) and the CERCA Programme/ Generalitat de Catalunya (to MC) and by the grant from Fondo Nacional de Desarrollo Científico y Tecnológico, FONDECYT 1240623 (to EM). FD-C is a recipient of a fellowship from the Agencia Nacional de Investigación y Desarrollo (ANID), Doctorado Nacional 21210611, Chile, and Scientific Exchange Grant 10594 from the European Molecular Biology Organization (EMBO), Germany. This work was carried out in part at the Esther Koplowitz Centre, Barcelona. Figures were created using BioRender (BioRender.com).

## Author contributions

**Francisco Díaz-Castro**: Conceptualization; Visualization; Writing—original draft; Writing—review and editing. **Eugenia Morselli**: Conceptualization; Visualization; Writing—original draft; Writing—review and editing. **Marc Claret**: Conceptualization; Visualization; Writing—original draft; Writing—review and editing.

## Disclosure and competing interests statement

The authors declare no competing interests.

