## [Peer Review File · EMBO Reports]

Interplay between the brain and adipose tissue: a metabolic conversation

Francisco Díaz-Castro, Eugenia Morselli, and Marc Claret

Corresponding author(s): Marc^{oo} Claret (mclaret@recerca.clinic.cat) , Eugenia Morselli (eugenia.morselli@uss.cl)

Review Timeline:

Submission Date:	10th Sep 24
Editorial Decision:	22nd Oct 24
Revision Received:	5th Nov 24
Accepted:	6th Nov 24

Editor: Deniz Senyilmaz Tiebe

Transaction Report:

Dear Marc,

Thank you again for the submission of your exciting review article to our editorial offices, which was indeed a pleasure to read! I have now received two referee reports that are copied below. As you will see, both referees state that your manuscript is interesting and timely. However, they have several suggestions to improve the submission that I kindly ask you to address in a revised manuscript, with the understanding that all referee points need to be addressed in a detailed point-by-point response.

I think it would be great if we can include your review in the last issue of 2024 of EMBO Reports. To be able to do that, the revised version needs to be submitted by October 31st the latest. Given the nature of the revisions required, it feels feasible to me. Please do contact me if you foresee problems meeting this deadline.

I further have these editorial requests:

- I would like to propose some minor changes in the abstract. Please confirm, or feel free to propose further changes:

The central nervous system and adipose tissue interact through complex communication. This bidirectional signaling regulates metabolic functions. The hypothalamus, a key homeostatic brain region, integrates exteroceptive and interoceptive signals to control appetite, energy expenditure, glucose, and lipid metabolism. This regulation is partly achieved via the nervous modulation of white (WAT) and brown (BAT) adipose tissue. In this review, we highlight the roles of sympathetic and parasympathetic innervation in regulating WAT and BAT activities, such as lipolysis and thermogenesis. Adipose tissue, in turn, plays a dual role as an energy reservoir and an endocrine organ, secreting hormones that influence brain function and metabolic health. Additionally, this review focuses on recently uncovered communication pathways, including extracellular vesicles and neuro-mesenchymal units, which add new layers of regulation and complexity to the brain-adipose tissue interaction. Finally, we also examine the consequences of disrupted communication between the brain and adipose tissue in metabolic disorders like obesity and type-2 diabetes, emphasizing the potential for new therapeutic strategies targeting these pathways to improve metabolic health.

- I think the figures look great. However, I also think that the font sizes of the figure labels need to be increased (especially Figure 1) to be readable upon resizing according to the publication format.

- Please provide 3-5 keywords for your review. These will be visible in the html version of the paper and on PubMed and will help increase the discoverability of your work.

- Please move the Disclosure and competing interest statement after Acknowledgments section.

- Please include the Figure Legends in the manuscript text (currently it is supplied as a separate file) and place it after References.

- We note that Figure 3 is currently not called out in the text.

- Author contributions section should be removed from the manuscript text.

- As per our format, the numbers of the section and the subsection titles need to be removed.

When submitting your revised manuscript, we will require a Microsoft Word file (.doc) of the revised manuscript text including detailed figure legends (at the very end), but without the figures.

Please provide the final figures as separate, high resolution files as .pdf, .eps, .tif, or .jpg (one file per figure). Please finalize the drafts provided and make sure they accurately illustrate the key scientific concepts that you wish to show.

Please also note the following points:

- If there are certain aspects of your figure draft that are based upon assumptions or where the scientific data remains ambiguous (for example, schematically depicting a presumed direct protein-protein interaction, protein shape or subcellular localizations etc.) please add a comment so that we can work with you on an accurate depiction. Please ensure the directionality and nature of interactions is presented accurately.

- If the figure or single panels of the figure have been adapted from a published figure, please add this information to the figure legend (e.g., 'Adapted from...' or 'Based on...'). The editor will discuss if a reference and permission will be necessary.

- Please only re-use figures or parts of a figure if this is essential for understanding the concept communicated. Often a reference to a previous paper will suffice. If the figure contains re-used images or elements of images, including schematics, micrographs or photos, please make sure that you have the permission/license to publish it (this also applies to your own previous work, if the journal you published in retains copyright. Certain 'creative commons' open access licenses, such as CC-BY 4.0, allow re-use without additional formal permissions). All re-used material must be explicitly cited.

- If you use an image data base for scientific iconography (e.g., BioRender), please let us know if you have a license that allows for publication in an academic journal. Often authors use misleading iconography for expedience. Please ensure the information

shown is scientifically accurate. If in doubt, please discuss with the editor.

I look forward to seeing a revised version of your manuscript when it is ready. Please let me know if you have questions or comments regarding the revision.

Kind regards,

Deniz

Deniz Senyilmaz Tiebe, PhD
Senior Scientific Editor
EMBO Reports

Referee #1:

In this perspective article Claret and colleagues assess recent knowledge on brain-adipose tissue crosstalk. Besides the "classical" view, based on endocrine and autonomic control of BAT and WAT, novel communications pathways, such as EVs and neuro-mesenchymal units are discussed.

This manuscript deals with a hot topic of the field, it is well written, and the authors have done an outstanding and authoritative job covering a large and broad range of information in a relatively concise manner. They also offer discussion about the physiological and pathophysiological relevance of the described findings, which adds substantial value to the manuscript. Overall, this is an excellent review article, and I only have some minor suggestions.

1. Considering the relevance of central actions of thyroid hormones and their key role in regulating BAT thermogenesis and browning on WAT, I would suggest some discussion of that.
2. In Figure 2 there is too much focus on the ARC and the PVH as the main efference of AgRP/NPY and POMC neurons. Other hypothalamic nuclei (such as the VMH) and areas (such the LHA), as well as central extra-hypothalamic sites (such as the raphe and the inferior oliva) could be added. This is important, because while afferent information is nicely explained (DRG neuron etc.), efferent information is over-simplified.
3. In Figure 3 important mechanisms, such as ER stress, autophagy and mitochondrial dysfunction are shown. However, there is no mention either to AMPK signaling (which is deeply commented in the text) or complex lipid metabolism; both could be included. In this sense, the role of hypothalamic ceramides as integrators of endocrine cues and SNS output to both BAT and WAT could be also discussed, in the main text and added to this Figure.

Referee #2:

This review provides a thorough and well written summary of the crosstalk between adipose tissue and the brain. The manuscript nicely details the current state of knowledge, while also describing emerging areas such as neuro-mesenchymal units and extracellular vesicles. This review will be accessible and informative for those not working in this field, but will also serve as a useful reference for researchers in this area. The following suggestions would further strengthen this manuscript:

1. On page 3, section 2.1, the authors state: "Each one of these fat pads is characterized by a distinct innervation density, specific receptor levels, and sympathetic nerve activity, which, in general, are higher in visceral than in subcutaneous depots (Willows et al., 2023a; Ibrahim, 2010)." In mouse models, many publications actually show higher sympathetic innervation in subcutaneous relative to visceral depots.
2. On page 3, section 2.1, the authors write: "This observation was also confirmed by advanced WAT clearing techniques, which allowed imaging of entire fat pads, revealing that 98.8% of presynaptic fibers within WAT are sympathetic (tyrosine hydroxylase (TH)-positive) (Jiang et al., 2017)." The authors should make it clear that sensory neurons can also stain for TH, so it would be incorrect to conclude that virtually all of the fibers are sympathetic.
3. Numerous references cited are review articles. Wherever possible, it would be preferable to cite primary literature.
4. The figures are informative, but somewhat rudimentary. More professionally designed figures would be nice to include in the published version.

REPONSE LETTER: Díaz-Castro et al (EMBOR-2024-60362V1)

Please find enclosed a revised version of our manuscript EMBOR-2024-60362V1 entitled "*Interplay between the brain and adipose tissue: a metabolic conversation*" by Díaz-Castro et al. that addresses the editorial and reviewer comments. We would like to thank our reviewers and the editorial team for both their constructive comments and insightful suggestions, which have been instrumental in strengthening the present manuscript. Our specific response to each of the comments is outlined below. **Please, note that additions in the new manuscript are highlighted with red font.**

EDITORIAL REQUESTS:

- I would like to propose some minor changes in the abstract. Please confirm, or feel free to propose further changes:

"The central nervous system and adipose tissue interact through complex communication. This bidirectional signaling regulates metabolic functions. The hypothalamus, a key homeostatic brain region, integrates exteroceptive and interoceptive signals to control appetite, energy expenditure, glucose, and lipid metabolism. This regulation is partly achieved via the nervous modulation of white (WAT) and brown (BAT) adipose tissue. In this review, we highlight the roles of sympathetic and parasympathetic innervation in regulating WAT and BAT activities, such as lipolysis and thermogenesis. Adipose tissue, in turn, plays a dual role as an energy reservoir and an endocrine organ, secreting hormones that influence brain function and metabolic health. Additionally, this review focuses on recently uncovered communication pathways, including extracellular vesicles and neuro-mesenchymal units, which add new layers of regulation and complexity to the brain-adipose tissue interaction. Finally, we also examine the consequences of disrupted communication between the brain and adipose tissue in metabolic disorders like obesity and type-2 diabetes, emphasizing the potential for new therapeutic strategies targeting these pathways to improve metabolic health".

We agree with the modifications and the new abstract has been added in the main text of the manuscript.

- I think the figures look great. However, I also think that the font sizes of the figure labels need to be increased (especially Figure 1) to be readable upon resizing according to the publication format.

We are pleased to hear that the figures meet your expectations. We have increased the font sizes as requested.

- Please provide 3-5 keywords for your review. These will be visible in the html version of the paper and on PubMed and will help increase the discoverability of your work.

Brain, brown adipose tissue, white adipose tissue, sympathetic and parasympathetic innervation, adipokines.

- Please move the Disclosure and competing interest statement after Acknowledgments section.

We have moved the Disclosure and Competing Interest statement to follow the Acknowledgments section as requested.

- Please include the Figure Legends in the manuscript text (currently it is supplied as a separate file) and place it after References.

The "Figure Legends" have been moved in the manuscript file, after References as requested.

- We note that Figure 3 is currently not called out in the text.

We apologize for this. Figure 3 has been now cited in the section "Consequences of brain-adipose tissue axis dysfunction" (page 18).

- Author contributions section should be removed from the manuscript text.

This section has been removed.

- As per our format, the numbers of the section and the subsection titles need to be removed.

The numbers of the sections have been removed.

Referee #1:

In this perspective article Claret and colleagues assess recent knowledge on brain-adipose tissue crosstalk. Besides the "classical" view, based on endocrine and autonomic control of BAT and WAT, novel communications pathways, such as EVs and neuro-mesenchymal units are discussed. This manuscript deals with a hot topic of the field, it is well written, and the authors have done an outstanding and authoritative job covering a large and broad range of information in a relatively concise manner. They also offer discussion about the physiological and pathophysiological relevance of the described findings, which adds substantial value to the manuscript. Overall, this is an excellent review article, and I only have some minor suggestions.

Thank you very much for your thoughtful review and positive feedback on our manuscript. We appreciate your acknowledgment of the comprehensive coverage and the added value of the physiological and pathophysiological aspects.

1. Considering the relevance of central actions of thyroid hormones and their key role in regulating BAT thermogenesis and browning on WAT, I would suggest some discussion of that.

We agree with the Reviewer that thyroid hormones play a key role in regulating BAT thermogenesis and browning of WAT. As such, we have added a brief discussion of the published data on that topic in the sections “Brain → White adipose tissue crosstalk” (page 5) and “Brain → Brown adipose tissue crosstalk; efferent innervation” (page 6-7).

2. In Figure 2 there is too much focus on the ARC and the PVH as the main efference of AgRP/NPY and POMC neurons. Other hypothalamic nuclei (such as the VMH) and areas (such as the LHA), as well as central extra-hypothalamic sites (such as the raphe and the inferior oliva) could be added. This is important, because while afferent information is nicely explained (DRG neuron etc.), efferent information is over-simplified.

We appreciate your suggestion to incorporate additional hypothalamic and extrahypothalamic nuclei in Figure 2, to offer a more comprehensive overview of the efferent pathways. In this regard, we have revised the figure to reflect this broader perspective. Additionally, we have modified the text to align with the updated content of the figure (page 7), ensuring a more balanced representation of both afferent and efferent information.

3. In Figure 3 important mechanisms, such as ER stress, autophagy and mitochondrial dysfunction are shown. However, there is no mention either to AMPK signaling (which is deeply commented in the text) or complex lipid metabolism; both could be included. In this sense, the role of hypothalamic ceramides as integrators of endocrine cues and SNS output to both BAT and WAT could be also discussed, in the main text and added to this Figure.

We thank the reviewer for this comment. Figure 3 illustrates the content of the section entitled “Consequences of Brain-Adipose Tissue Axis Dysfunction.” In this section, we focus on some of the primary pathophysiological molecular processes occurring in the hypothalamus in the context of metabolic dysfunction, without emphasizing any specific signaling pathways. As such, Figure 3 is designed to reflect this broader biological perspective rather than detailing individual pathways like AMPK. While AMPK is indeed a critical signaling node involved in various functions, it was not specifically referenced in this section. Therefore, we believe that the decision to exclude AMPK from Figure 3 is justified, as our approach aims to enhance the understanding of the complex interactions at play in brain-adipose tissue axis dysfunction at the process level.

Regarding the role of ceramides, we agree with the reviewer that this is an important addition to the review. In response, we have incorporated a brief discussion on the role of ceramides in the section “Brain → Brown Adipose Tissue Crosstalk: Metabolic Effects via Efferent Innervation” (page 8), as well as in the section “Consequences of Brain-Adipose Tissue Axis Dysfunction” (page 19). These sections detail how ceramide signaling in the brain affects sympathetic outflow to BAT, modulating thermogenesis and energy balance, and further explore the impact of dysregulated brain-adipose communication on systemic metabolism.

Referee #2:

This review provides a thorough and well written summary of the crosstalk between adipose tissue and the brain. The manuscript nicely details the current state of knowledge, while also describing emerging areas such as neuro-mesenchymal units and extracellular vesicles. This review will be accessible and informative for those not working in this field, but will also serve as a useful reference for researchers in this area.

Thank you for your positive feedback on our review.

The following suggestions would further strengthen this manuscript:

1. On page 3, section 2.1, the authors state: "Each one of these fat pads is characterized by a distinct innervation density, specific receptor levels, and sympathetic nerve activity, which, in general, are higher in visceral than in subcutaneous depots (Willows et al., 2023a; Ibrahim, 2010)." In mouse models, many publications actually show higher sympathetic innervation in subcutaneous relative to visceral depots.

We thank this Reviewer for the observation. After reviewing the literature, we identified studies indicating higher sympathetic innervation in subcutaneous depots compared to visceral ones (PMID: 24480348), as well as studies reporting the opposite (PMID: 24452544), depending on the mouse model and experimental conditions used. Given these discrepancies, we have decided to choose a neutral position and remove the specific sentence referring to this on page 3 for clarity. The revised sentence now reads: *"Each one of these fat pads is characterized by a distinct innervation density, specific receptor levels, and sympathetic nerve activity (Willows et al, 2023a; Vaughan et al, 2014)."*

2. On page 3, section 2.1, the authors write: "This observation was also confirmed by advanced WAT clearing techniques, which allowed imaging of entire fat pads, revealing that 98.8% of presynaptic fibers within WAT are sympathetic (tyrosine hydroxylase (TH)-positive) (Jiang et al., 2017)." The authors should make it clear that sensory neurons can also stain for TH, so it would be incorrect to conclude that virtually all of the fibers are sympathetic.

We thank this Reviewer for the observation. We revised the cited paper in this sentence and the authors performed a co-immunolabeling of synaptophysin and TH. As such they concluded that intra-adipose neural arborizations are predominantly sympathetic inputs. This sentence (page 3) has been modified accordingly to provide a more nuanced statement. The revised sentence now reads: *"This observation was also confirmed by advanced WAT clearing techniques, which allowed imaging of entire fat pads, revealing that almost 99% of presynaptic fibers within WAT are likely sympathetic (tyrosine hydroxylase (TH) and synaptophysin positive) (Jiang et al, 2017)."*

3. Numerous references cited are review articles. Wherever possible, it would be preferable to cite primary literature.

We appreciate this comment. As requested, we have reviewed all the cited articles and replaced review articles with original contributions wherever possible. In general, our criteria were to retain review articles for general statements while incorporating original references for more specific information. Additionally, we have also identified redundant review articles. In these instances, we chose to retain only the most relevant and/or more recent reviews that effectively support the statements in the text. Below is a list of the modifications made to the reference list:

- Collins 2011: excluded (redundant).
- Ibrahim, 2010: excluded (biased statement on the sympathetic nerve activity in scWAT vs vWAT).
- Arner, 2005: excluded (redundant) and replaced by Lass et al. (2011).
- Ceddia & Collins, 2020: excluded (redundant).
- Richard & Picard, 2011: excluded (redundant).
- Klingenberg, 1990: excluded (redundant) and replaced by Cannon & Nedergaard (2004).
- Labbé et al., 2015: excluded (redundant) and replaced by Contreras et al (2017).
- Schwartz *et al.*, 2000: excluded (redundant).
- Liu et al., 2020: excluded (redundant).
- Richard et al., 2009: excluded and replaced by Verty et al 2011.
- Pan & Myers, 2018: excluded (redundant).
- Mathew et al., 2018: excluded (redundant).
- Le Foll, 2019: excluded (redundant).
- Freire et al., 2022: replaced by PMID: 33774223.
- Christie et al., 2018: replaced by PMID: 9349813.
- Saito et al., 2020: excluded (redundant).
- Villarroya et al., 2017: excluded (redundant) and replaced by Ahmad et al (2021).
- Yang & Stanford, 2022: excluded (redundant) and replaced by Ahmad et al (2021).
- Liu et al., 2023: excluded (redundant) and replaced by Isaac et al (2021).
- Bond et al., 2022: excluded (redundant) and replaced by Isaac et al (2021).
- Kwan et al., 2021: excluded (redundant)

4. The figures are informative, but somewhat rudimentary. More professionally designed figures would be nice to include in the published version.

Thank you for your feedback regarding the figures. We have made some adjustments in the Figures based on the comments from the editor and Reviewer #1. We hope that these changes enhance the clarity and effectiveness of the illustrations.

POINT BY POINT LIST OF ALTERATIONS

Page 1: new abstract added (as requested by the editor).

Page 1: key words added (as requested by the editor).

Page 3: nuanced statement on sensory neurons can also stain for TH (requested by Reviewer 2).

Page 5: addition on thyroid hormones (requested by Reviewer 1).

Page 6 and 7: addition on thyroid hormones (requested by Reviewer 1).

Page 7: addition on extrahypothalamic sites involved in thermogenesis control (requested by Reviewer 1).

Page 8: addition on the role of ceramides in thermoregulation (requested by Reviewer 1).

Page 9: nuanced statement on cannabinoid signaling.

Page 18: reference to Figure 3 (requested by editor).

Page 19: addition on the role of ceramides in pathophysiology (requested by Reviewer 1).

Page 20: disclosure and Competing Interest statement to follow the Acknowledgments section as requested by the editor.

Pages 21-38 (References): To support the new text additions, we have incorporated several new references to strengthen our statements. These include:

- Contreras et al 2014 (PMID: 25284795)
- Dib et al 1994 (PMID: 7953666)
- González-García et al 2017 (PMID: 27728904)
- Hale et al 2011 (PMID: 21111735)
- Higgins et al 1988 (PMID: 2467224)
- Schneeberger et al 2019 (PMID: 31257028)
- Uno & Shibita 2001 (PMID: 11208585)
- Verty et al 2009 (PMID: 19057531)
- Verty et al 2011 (PMID: 21412227)
- Weiss & Aghajanian 1971 (DOI: 10.1016/0006-8993(71)90540-3)

Page 39-40: Figure Legends added (requested by the editor).

Please note that the author contributions section and numbers in sections have been removed as requested by the editor.

Dr. Marc Claret
IDIBAPS
Rosselló 149-153
Barcelona, Barcelona 08036
Spain

Dear Marc,

Thank you for submitting your revised review. I am very pleased to inform you that your review has been accepted for publication in EMBO Reports.

Your manuscript will be processed for publication by EMBO Press. It will be copy edited and you will receive page proofs prior to publication.

You will soon be contacted by Springer Nature to sign your publishing license. When you login to the customer service website, please use the following token to waive the article publication charges: MJAWODIXNJYWOQ

Should you experience any difficulty, please email publishing@embo.org.

If you have any questions, please do not hesitate to contact the Editorial Office. Thank you very much for your contribution to EMBO Reports.

Kind regards,

Deniz

Deniz Senyilmaz Tiebe, PhD
Senior Scientific Editor
EMBO Reports
